# NUP214 in Leukemia: It’s More than Transport

**DOI:** 10.3390/cells8010076

**Published:** 2019-01-21

**Authors:** Adélia Mendes, Birthe Fahrenkrog

**Affiliations:** Institute of Biology and Molecular Medicine, Université Libre de Bruxelles, 6041 Charleroi, Belgium; mrodrig6@ulb.ac.be

**Keywords:** NUP214, SET, DEK, leukemia, HOX genes, CRM1, nucleocytoplasmic transport, chromatin remodeling, transcription

## Abstract

NUP214 is a component of the nuclear pore complex (NPC) with a key role in protein and mRNA nuclear export. Chromosomal translocations involving the *NUP214* locus are recurrent in acute leukemia and frequently fuse the C-terminal region of NUP214 with SET and DEK, two chromatin remodeling proteins with roles in transcription regulation. SET-NUP214 and DEK-NUP214 fusion proteins disrupt protein nuclear export by inhibition of the nuclear export receptor CRM1, which results in the aberrant accumulation of CRM1 protein cargoes in the nucleus. SET-NUP214 is primarily associated with acute lymphoblastic leukemia (ALL), whereas DEK-NUP214 exclusively results in acute myeloid leukemia (AML), indicating different leukemogenic driver mechanisms. Secondary mutations in leukemic blasts may contribute to the different leukemia outcomes. Additional layers of complexity arise from the respective functions of SET and DEK in transcription regulation and chromatin remodeling, which may drive malignant hematopoietic transformation more towards ALL or AML. Another, less frequent fusion protein involving the C terminus of NUP214 results in the sequestosome-1 (SQSTM1)-NUP214 chimera, which was detected in ALL. SQSTM1 is a ubiquitin-binding protein required for proper autophagy induction, linking the NUP214 fusion protein to yet another cellular mechanism. The scope of this review is to summarize the general features of NUP214-related leukemia and discuss how distinct chromosomal translocation partners can influence the cellular effects of NUP214 fusion proteins in leukemia.

## 1. Introduction

The hematopoietic compartment is, not seldom, home for development of neoplastic malignancies. In 2012, more than 80,000 people in Europe were diagnosed with leukemia and more than 50,000 deaths were registered in the same year [1]. Leukemia comprises a wide and heterogeneous group of neoplasms that affect the production and the maturation of white blood cells. It can be roughly divided into two major classes: chronic and acute leukemia. While chronic leukemia is slow-growing, the latter refers to a rapid increase in the proliferation of bone marrow progenitor cells (BMPCs) that are blocked in early stages of cell differentiation [2]. Both chronic and acute leukemia are categorized as myelogenous/myeloid and lymphocytic/lymphoblastic.

Acute myeloid leukemia (AML) is the most common form of acute leukemia in adults and the second most common in children. AML is generally considered a disease of the elderly, with a median age at diagnosis of 67 years. Its incidence and patient mortality rates increase with age (www.seer.cancer.gov). The most aggressive forms of AML are, however, diagnosed in children and young adults [3]. Acute lymphoblastic leukemia (ALL) is the prevalent form of childhood leukemia and the second most common in adults [4]. Like AML, ALL is highly aggressive and characterized by rapid progression. Proximal causes that lead to the development of leukemia can be categorized into three major groups: (1) gene mutations and translocations, (2) dysregulation of the immune system, and (3) changes in the bone marrow environment [5]. Chromosomal translocations result in the expression of proto-oncogenic fusion proteins that, dependent on the particular nature of the fusion protein, strongly impact disease progression and patient prognosis.

Frequent targets for chromosomal translocations are the nucleoporin genes *NUP214* and *NUP98*. Nucleoporins are integral parts of the nuclear pore complex (NPC), the sole communication gateway between the two major cellular compartments: the nucleus and the cytoplasm (Figure 1). Each NPC is structurally a roughly tripartite protein complex that is composed of multiple copies of its about 30 molecular units, the nucleoporins [6,7,8]. NPCs are anchored in the nuclear envelope (NE) by their central scaffold (also known as central framework or spoke ring complex), which is flanked by two ring moieties, the nuclear and the cytoplasmic ring. Filamentous structures known as nuclear basket and cytoplasmic filaments project from both ring moieties into the respective compartment [6,9].

NPCs define the permeability of the NE in two ways: they restrict diffusion to molecules <40 kDa and they mediate selective nucleocytoplasmic transport, which is amongst others essential for the rapid adaptation of cells to stress conditions, such as genotoxic and oxidative stress or hypoxia [10,11,12]. Critical for nucleocytoplasmic transport are repetitive stretches of phenylalanine-glycine (FG) motifs found in about one third of the nucleoporins. Individual FG motifs are interspersed with charged residues and are enriched in disorder promoting amino acids. Consequently, FG domains lack a secondary structure and are considered as natively unfolded [13,14]. These FG domains contribute to the dual character of NPCs: they allow selective transport of certain proteins and RNAs, while acting as selective barrier to others [13,15]. Selective transport through the NPC is mediated by transport receptors from the karyopherin-β family of proteins, which establish multiple dynamic interactions with the FG repeats in the nucleoporins to ferry their cargo across the NPC [16,17,18,19].

Karyopherin-β proteins are distinguished as importins and exportins. The major nuclear export receptor is chromosomal maintenance 1 (CRM1; also known as exportin 1/XPO1). CRM1 recognizes proteins that carry one or more leucine-rich nuclear export signals (NES) and mediates their nuclear export [20,21,22]. To ferry NES-cargoes from the nucleus into the cytoplasm, CRM1 uses nuclear RanGTP as a cofactor for the formation of export complexes [23]. Upon NPC-mediated export, the disassembly of nuclear export complexes depends on GTP hydrolysis by the Ran GTPase-activating protein 1 (RanGAP1) and its cofactors Ran-binding proteins 1 and 2 (RanBP1, RanBP2), which generates RanGDP [21,24,25]. In the cytoplasm, the nuclear transport factor 2 (NTF2) binds to RanGDP and mediates its import back into the nucleus, where the Ran nucleotide exchange factor regulator of chromosome condensation 1 (RCC1) promotes the replacement of Ran-bound GDP by GTP to establish RanGTP. The compartmentalization of the Ran cofactors ensures the maintenance of a RanGDP/GTP gradient throughout the cell, which is essential for continuous nucleocytoplasmic transport [23,26].

NUP214 and NUP98 are two FG nucleoporins with crucial roles in nucleocytoplasmic transport as they interact with multiple transport receptors of the karyopherin-β family and mediate their passage through the NPC [27,28,29,30,31]. Apart from their NPC-related function, these two nucleoporins are involved in other cellular processes, such as cell cycle regulation, mitosis and gene expression [32,33,34,35,36,37,38]. Chromosomal translocations involving the *NUP214* and *NUP98* loci are recurrent in AML and ALL. They may occur as a consequence of previous cancer therapy, but also de novo. To date, 30 different partners for NUP98 are known, which fall into two main categories: homeodomain (HD) or non-HD chromatin binding proteins [39,40,41]. Arrest of cellular differentiation and upregulation of clustered *HOX* genes are common to all NUP98 fusion proteins [42,43,44,45]. Leukemogenic NUP98 fusion proteins have been extensively studied and NUP98-related leukemia has been addressed in several review articles in recent years [41,42,43,44,46,47,48,49]. The biological effects and molecular mechanisms associated with NUP214 fusion proteins are in contrast less well characterized. Compared to other leukemia subtypes, NUP214-associated leukemia is highly aggressive and patients are frequently refractory to treatment, which coincides with overall poorer survival rates [50,51,52,53]. Additionally, patients with NUP214 leukemia often present secondary mutations that influence the course of disease [50,54]. Targeted therapies may significantly improve therapy outcome for patients suffering from NUP214 chromosomal rearrangements. To allow the development of such new specific targeted therapies an in-depth understanding of the impact of NUP214 fusion proteins on normal cellular behavior will be key. To arrive at this, the normal functions of NUP214 and its fusion partners need to be unraveled and understood. The current state of knowledge in respect thereof will be discussed in this review article.

## 2. NUP214 Is Critical for Nucleocytoplasmic Transport

NUP214 is an FG nucleoporin anchored to the cytoplasmic ring of the NPC and forms a subcomplex with nucleoporin NUP88 (Figure 1) [27,55,56]. Structurally, NUP214 is composed of three domains: a N-terminal β-propeller domain, two central coiled-coil motifs that mediate the interaction with NUP88 and anchor NUP214 to the NPC, and a C-terminal FxFG domain that can be detected on both sides of the NPC (Figure 2) [55,57].

A role for NUP214 in nucleocytoplasmic transport was early established by studies in mice and in human cells: depletion and overexpression of NUP214 either resulted in the nuclear accumulation of proteins and poly(A^+^) RNA [32,33]. In this context, NUP214 is known to interact with both CRM1, the major exportin for proteins, and with nuclear RNA export factor 1 (NXF1), the principal mRNA export factor [27,28,58,59]. NUP214 exhibits multiple binding sites for CRM1 in its FG domain and, among all nucleoporins, the highest affinity for CRM1 [19,60]. The interaction between the FG domain of NUP214 (NUP214_FG_) and CRM1 induces conformational changes in both proteins and promotes the stabilization of CRM1-export complexes at the cytoplasmic filaments of the NPC [60]. NUP214 acts as a final anchoring site immediately before the disassembly of CRM1 export complexes and release of the cargoes into the cytoplasm [19,60].

The role of NUP214 in mRNA export from the nucleus is less well understood. NUP214 interacts, via its N-terminal region, with NXF1 and the ATP-dependent DEAD-box helicase 19 (DDX19) [58,61,62]. NUP214 stabilizes DDX19’s localization at the cytoplasmic periphery of the NPC. Together with GLE1, another important mRNA export factor, it regulates DDX19’s ATPase activity as well as mRNP disassembly in the cytoplasm [63]. It has been recently suggested that NUP214 can function both as inhibitor and stimulator of DDX19 activity, however the exact role of NUP214 in the trafficking of mRNA remains to be established [63].

Deregulation of NUP214 protein levels not only affects nucleocytoplasmic transport, but also results in cell cycle arrest and mitotic aberrations, whereas NPC architecture is not affected [32,33]. Moreover, genomic knockout of *Nup214* led to embryonic lethality in mice [32,33,64].

## 3. NUP214 Is a Recurrent Player in Acute Leukemia

Chromosomal translocations involving the *NUP214* locus are a recurrent event in acute leukemia [50,52,53,54]. To date, four distinct NUP214 fusion proteins have been described (Table 1). Initially, NUP214 was mapped in chromosomal rearrangements that led to its fusion with two distinct chromatin binding proteins, DEK and SET. While DEK-NUP214 (t(6;9)(p22;q34)) was associated with AML, SET-NUP214 (del(9)(q34.11q34.13)) came along with ALL, and has been reported sporadically in acute undifferentiated leukemia (AUL) [65,66,67]. Other NUP214 fusion proteins are sequestosome-1(SQSTM1)-NUP214 (der(5)t(5;9)(q35;q34)) and NUP214-ABL1, which results from the episomal amplification of chromosome arm 9q. SQSTM1-NUP214 and NUP214-ABL1 were both identified in T-cell ALL (T-ALL) patients [68,69]. Additionally, two *NUP214* fusion transcripts have been reported: the *NUP214-RAC1* transcript was detected in a patient carrying DEK-NUP214 and the *NUP214-XKR3* transcript was observed in the CML cell line K562 that expresses the BCR-ABL1 fusion protein [70,71]. However, neither of these transcripts was detected at the protein level. *NUP214* chromosomal translocations, with the exception of NUP214-ABL1, are associated with an aggressive course of disease and poor chance of survival [50,51,52,53].

Out of the four mentioned NUP214 fusion proteins, three retain their FG domain. While SET-NUP214 and DEK-NUP214 conserve virtually the entire NUP214_FG_ domain (42 out of 44 FG repeats), SQSTM1-NUP214 preserves 14 FG repeats [65,66,67,69,72]. In SET-NUP214 and DEK-NUP214, the NUP214 portion is C-terminally fused with almost the entire peptide sequence of the respective partner (Figure 2) [67,72]. Given the presence of several binding sites for CRM1 in NUP214’s FG domain, the preservation of NUP214_FG_ in the fusion proteins suggests that CRM1-mediated nuclear export may be affected in cells carrying *NUP214* chromosomal translocations. Recent reports indeed demonstrated that leukemogenic NUP214 fusion proteins can impair nuclear export by sequestering CRM1 into nuclear bodies [73,74]. It remains, however, to be clarified whether NUP214 fusion proteins affect CRM1-mediated export in general or only a subset of cargoes and how this is promoting malignant transformation during hematopoietic development.

A further common characteristic of SET-NUP214, DEK-NUP214, and SQSTM1-NUP214: they all promote upregulation of *HOX* genes, namely *HOXA6-10* and the entire *HOXB* cluster, with the exception of *HOXB7* and *HOXB13* [39,50,54,75,76]. It was recently demonstrated that *HOX* promoter regions are enriched in CRM1 [77]. Interestingly, this chromatin-bound CRM1 can recruit the AML-associated NUP98-HOXA9 fusion protein. NUP98-HOXA9 binds CRM1 on chromatin via the FG domain of NUP98 (NUP98_FG_) and attracts epigenetic regulators, which leads to aberrant activation of *HOX* gene expression [77]. It is hence tempting to speculate that NUP214 fusion proteins use a similar molecular mechanism to induce upregulation of *HOX* gene expression. Whether this is indeed the case remains to be seen.

## 4. SET-NUP214 and DEK-NUP214: Same, Same, but Different

### 4.1. SET-NUP214 and DEK-NUP214 Act via Shared Molecular Mechanisms

The observation that *SET-NUP214* and *DEK-NUP214* have the same genomic breakpoint in *NUP214* and therefore conserve the same portion of NUP214, i.e., NUP214_FG_, was an early indication that NUP214_FG_ is important for the transformation potential of both fusion proteins [65,66,67,72,73,74]. Recent studies highlighted common features between SET-NUP214 and DEK-NUP214 in cells overexpressing the particular fusion proteins. SET-NUP214 and DEK-NUP214 both accumulated in nucleoplasmic foci, despite differences in size and frequency. Endogenous CRM1 and its known cargoes RanBP1 and HIV-1 Rev reporter cargoes are sequestered into these foci, which leads to their nuclear accumulation due to impaired CRM1 activity [73,74]. Additionally, both fusion proteins can impair NF-ĸB-mediated transcription in a CRM1-dependent manner [74]. It remains to be clarified whether NUP214 fusion proteins target general protein nuclear export or specific CRM1 cargoes.

Supporting the view that SET-NUP214 and DEK-NUP214 act via shared molecular mechanisms is the fact that SET and DEK are both histone binding proteins and epigenetic regulators [78,79,80,81,82,83,84,85,86]. SET and DEK hinder chromatin relaxation by inhibition of the histone acetyltransferases (HATs) p300 and CREB-binding protein (p300/CBP) and the p300/CBP associated factor (PCAF; also known as acetyltransferase 2B/KAT2B). They act as negative regulators of transcription in this context [78,83,85,87,88,89]. The inhibitory activities of SET and DEK on HATs are mediated by their acidic domains that bind lysine residues in histones H3 and H4 [84,88]. NUP214 fusion proteins preserve the almost entire sequence of SET and DEK, including their histone binding domains (Figure 2) [67,72]. Therefore, it is possible that in the context of NUP214 fusion proteins, SET and DEK activity may be deregulated, for example due to altered association with chromatin when fused to NUP214_FG_, which in turn would lead to abnormal histone hypoacetylation. Considering that NUP214 leukemia targets early hematopoietic progenitors [90], aberrant hypoacetylation mediated by the fusion proteins could lead to the silencing of genes important for lineage commitment and hematopoietic differentiation.

### 4.2. SET-NUP214 and DEK-NUP214 Associate with Distinct Gene Expression Profiles

Patients with SET-NUP214 or DEK-NUP214 are comparatively young, with a median age of diagnosis of 27.5 and 23 years, respectively [52,54]. Due to fast disease progression and chemotherapy resistance, survival rates of the patients are low, independent of the respective leukemia subtype [50,52,54]. SET-NUP214 has been mostly reported in T-ALL and rarely in AML and AUL, while DEK-NUP214 has been exclusively reported in myelodysplastic syndrome (MDS) and AML, indicating that distinct cellular pathways drive leukemogenic progression in patients with either fusion transcript [50,54]. In fact, the presence of the *DEK-NUP214* transcript in AML patients is considered as an independent marker of unfavorable prognosis [91]. In transgenic mouse models, expression of SET-NUP214 induces the expansion of BMPCs and impairs cell differentiation, but it does not induce leukemia [92,93]. In contrast, DEK-NUP214 expression is sufficient to cause AML by targeting CD34^+^ hematopoietic progenitor cells [90]. However, in an immunocompetent mouse model, the expression of DEK-NUP214 resulted in leukemia induction with low efficiency, and only from a small subpopulation of hematopoietic stem cells [94]. In this mouse model DEK-NUP214 induced only a slight decrease in the differentiation of hematopoietic precursors [94]. This contrasts with the immunophenotypic findings in AML patients with DEK-NUP214 expression, showing a differentiation block in hematopoietic cells [52]. These findings support the idea that additional mutations are required to promote malignant transformation. Leukemia patients with SET-NUP214 or DEK-NUP214 exhibit distinct mutation profiles, which could explain the differences in disease progression [50,52,95]. In a series of adult T-ALL patients, the presence of SET-NUP214 correlated with upregulation of *HOX* genes and with high expression levels of the lymphoblastic leukemia associated hematopoiesis regulator 1(LYL1) and the myocyte enhancer factor 2C (MEF2C) genes [95]. SET-NUP214 was also associated with mutations of the PHD finger protein 6 (PHF6) and the neurogenic locus notch homolog protein 1 (NOTCH1) genes [95,96]. All these genes are involved in hematopoietic development and are frequently mutated in myeloid and lymphoblastic leukemia [96,97,98].

A distinct gene expression profile was revealed in pediatric AML patients harboring DEK-NUP214. In these cases, a significant upregulation of *EYA3*, a homolog of the *Drosophila* eyes absent gene, of *Sestrin1* (SESN1; a p53-regulated gene with a role in DNA damage response), of *PR domain 2* (PRDM2/RIZ; a member of a superfamily of histone/protein methyltransferases), and of *Histone cluster 1 H4 Family Member C* (HIST2H4) genes was observed [50]. Additionally, Fms-related tyrosine kinase 3 (FLT3) internal tandem duplication (FLT3-ITD), a frequent mutation in AML that coincides with poor prognosis, is in DEK-NUP214 patients higher than accidental occurrence [50,52]. Differences in gene expression profiles in patients indicate that SET-NUP214 and DEK-NUP214 may execute their functions in distinct cellular backgrounds, despite the fact that they also regulate similar molecular mechanisms. Besides secondary mutations, disease progression in SET-NUP214- or DEK-NUP214-associated leukemia is certainly also influenced by the function of SET and DEK in their respective chimeric proteins. Key to an ultimate comprehension of how NUP214 chimeras cause malignant transformation is to decipher the specific function of each fusion partner and how it can modulate or be modulated in the resulting fusion proteins.

## 5. The SET Protein

In eukaryotic cells, SET is expressed in two isoforms: TAF1-α and TAF1-β, which are generated by alternative splicing of the first two exons of the *TAF1* gene (www.uniprot.org, protein ID: Q01105). In SET-NUP214, only the TAF1-β isoform is present [66]. SET/TAF1-α and SET/TAF1-β are localized within the nucleus and have histone binding and chromatin remodeling activity [78,80,81,87,88,99]. Structurally, SET/TAF1-β is composed by a N-terminal dimerization domain, a central “earmuff” domain (so called due to its headphone-like structure), and a negatively charged C-terminal acidic domain (Figure 2) [88]. Both SET isoforms, together with the acidic nuclear phosphoprotein pp32, compose the inhibitor of acetyltransferase (INHAT) complex that negatively regulates histone acetylation mediated by p300/CBP and PCAF [78,79,100,101,102]. In the context of the INHAT complex, SET/TAF1-β and pp32 bind lysine residues of histone tails and inhibit their HAT-mediated acetylation by a mechanism called histone masking [78]. Both the earmuff and acidic domain (INHAT domains) are required for the INHAT activity of SET [78,88].

Additionally, SET/TAF1-β can regulate acetylation of non-histone proteins independent of the INHAT complex. However, this INHAT-independent function of SET/TAF1-β requires its INHAT-domains for the interaction with the target proteins [99,103,104,105,106]. In basal conditions, SET/TAF1-β binds directly to lysines of transcription factors, such as p53, FOXO1, and the ligand-activated transcription factor glucocorticoid receptor (GR), and negatively regulates their activity [103,104,105].

SET/TAF1-β inhibits GR signaling by interacting with GR and GR responsive elements (GREs) on target gene promoters [103]. In the presence of corticosteroids, which are GR activating ligands, the binding of SET/TAF1-β to GR and GREs is lost, which in turn leads to the transcription of GR target genes. SET-NUP214 interacts with GREs, but not with the receptor itself. Moreover, the interaction between GREs and SET-NUP214 is not disrupted by corticosteroids. Therefore, transcription of GR target genes is not initiated, despite activation of GR [103]. Corticosteroid therapy to induce transcription of GR target genes is a reference treatment in ALL [107]. However, expression of SET-NUP214 in ALL patients is highly correlated with resistance to corticosteroid treatment, possibly due to inappropriate transcriptional regulation of GR target genes by SET-NUP214 [108].

Upregulation of *HOXA* and *HOXB* clusters is a well-known event in SET-NUP214 leukemia, which results in failed cellular differentiation [76]. The mixed lineage leukemia (MLL) histone methyltransferase is an activator of *HOX* gene expression [109,110]. In MLL-rearranged leukemia, the recruitment of histone methyltransferase DOT1L, which mediates mono-, di-, and tri-methylation of lysine 79 of histone 3 (H3K79), by MLL fusion proteins is essential for the upregulation of *HOXA* genes [111]. An interaction between SET/TAF1-β and MLL has been reported in HeLa cells overexpressing both proteins and resulted in a synergistic effect on the activation of *HOXA9* expression [112]. In another report, the combined knockdown of SET and SET-NUP214 resulted in a reduction of *HOXA9* gene expression in a leukemia cell line with endogenous expression of the fusion protein. *HOXA9* inactivation was accompanied by a reduction in H3K79 methylation [76]. Expression of *HOXA* clustered genes was not affected by the specific knockdown of SET. This indicates that SET-NUP214 requires normal SET to activate *HOXA* gene expression, which was supported by the observation of an interaction between SET, SET-NUP214, and DOT1L [76]. Therefore, it is conceivable that SET-NUP214 recruits both SET and DOT1L to the promoter of *HOX* genes thereby causing their upregulation.

## 6. The DEK Protein

DEK is ubiquitously expressed and found mainly associated with chromatin by two distinct DNA binding domains: the central region of DEK, which includes a conserved sequence element called the scaffold attachment factor (SAF)-box and a DNA binding domain at the C terminus of DEK, in between acidic stretches (Figure 2; www.uniprot.org, protein ID: P35659) [83,113,114]. DEK is a key player in the maintenance of chromatin stability under normal and stress conditions [83,85,114]. It binds preferentially to supercoiled and four-way junction DNA and can introduce positive supercoils [83,114]. In embryonic stem cells (ESCs), DEK appears to regulate histone deposition onto chromatin and to preserve telomere integrity [115]. Via its acidic domains, DEK binds lysine residues on histones H3 and H4 and inhibits their HAT-mediated acetylation by p300/CBP and PCAF [84]. Unlike SET, the exact molecular mechanism leading to histone hypoacetylation is still unclear. However, DEK is part of a transcriptional repressor complex, which involves the death domain-associated protein 6 (DAXX6) and the histone deacetylase II (HDAC2) [82]. Therefore, the regulatory role of DEK in transcription may derive from the combined action between direct binding to histones and the recruitment of repressor proteins to gene promoters.

Studies on the role of DEK in transcription regulation have led to conflicting results [116]. Chromatin immunoprecipitation (ChIP) experiments revealed that DEK is mainly associated with euchromatin where it is enriched at transcription start sites of actively transcribed genes. However, in the same report, DEK was also observed in heterochromatin regions, where it was associated with repressive histone marks [86]. DEK depletion resulted in either upregulation or downregulation of its target genes, which favors a scenario in which DEK-mediated transcriptional regulation is determined by specific, yet unknown, factors [86,116]. Currently, it is not known if (or how) the histone binding functions of DEK are affected by DEK-NUP214. However, the fusion protein preserves the acidic domains of DEK, which are required for the binding of DEK to histone lysine residues (Figure 2) [67,117].

In support of its multifunctional roles, a recent study on the interactome of DEK in human cells identified hundreds of interactors, including chromatin remodeling, RNA processing, ribosome biogenesis, and stress response factors [118]. Gene ontology (GO) analysis further unveiled that proteins involved in translation were among the most frequent hits [118]. In this context, it was previously shown that DEK-NUP214 activates the mTOR signaling pathway and the eukaryotic translation initiation factor 4E (EIF4E), which coincided with increased protein production [119,120]. While the exact mechanism by which DEK-NUP214 regulates mTOR and EIF4 is not clear, it is conceivable that it enhances the interaction between DEK and translation factors. Protein levels of DEK itself are significantly increased in several solid cancers, such as colon cancer, hepatocarcinoma, and melanoma, and are associated with poor treatment outcome [121,122,123,124,125,126]. Studies of DEK protein levels in AML samples on the other hand revealed inconsistent results [127,128]. This suggests that despite the ubiquitous nature of DEK expression, its functions are probably cell and tissue specific [129].

During hematopoiesis, DEK promotes myeloid differentiation by interacting with the myeloid associated transcription factor CCAATT/enhancer-binding protein α (C/EBPα) [89]. Phosphorylation of C/EBPα by FLT3 inhibits the transcriptional activity of C/EBPα and results in a differentiation arrest of myeloid cells [89]. Constitutively active FLT3-ITD, which is frequently observed in DEK-NUP214 patient cells [89,130,131], repressed C/EBPα expression [132], suggesting that phosphorylation of C/EBPα impairs both its expression and its function. This in turn, may contribute to the differentiation block observed in DEK-NUP214 leukemia cells. Furthermore, in the absence of FLT3-ITD, DEK-NUP214 can block cellular differentiation of early hematopoietic progenitors by upregulation of *HOXA9/10* and *HOXB2/3* genes, by a yet unknown molecular mechanism [90].

Although yet unstudied in the context of leukemia, a role for DEK in DNA-double strand break (DSB) repair may contribute to the poor response of patients to chemotherapy. DEK was recently identified as a component of the ataxia-telangiectasia mutated (ATM)-dependent homologous recombination (HR) pathway [133,134]. In the presence of irradiation-induced DSBs, ATM phosphorylates the histone variant H2AX, which leads to its activation and is referred to as γ-H2AX [135]. γ-H2AX recruits the DNA repair protein RAD51 onto chromatin, in an apparently DEK-dependent manner [133]. Either the DSB repair pathway, non-homologous end joining (NHEJ) and HR, were impaired in cells depleted for DEK, which rendered them more sensitive to genotoxic stimuli [133]. Due to its ability to bind complex DNA structures, DEK facilitates cell proliferation under conditions of DNA replication stress by resolving complex DNA structures at stalled replication forks, which in turn promotes replication fork progression [136].

## 7. SQSTM1-NUP214: Playing with Autophagy

The rarest leukemic NUP214 fusion protein is SQSTM1-NUP214: only two patients, one with ALL and another with AML, have been reported so far [69,137]. SQSTM1-NUP214 comprises the N-terminal five exons of SQSTM1 fused with the partial C terminus of NUP214, including its final 14 FG repeats (Figure 2) [69]. The leukemogenic activity of SQSTM1-NUP214 is not well understood, although it was recently demonstrated that SQSTM1-NUP214 can interact with CRM1 and impair its function, similar to SET-NUP214 [73]. However, due to the shorter NUP214_FG_ fragment, SQSTM1-NUP214 induced less nuclear accumulation of proteins and polyA^+^ RNA in comparison to SET-NUP214 [73]. Transcriptome analysis further revealed that SQSTM1-NUP214 led to upregulation of clustered *HOX* genes (*HOXA3*, *HOXA5*, *HOXA7*, *HOXA9*, and *HOXA10*), as previously reported for SET-NUP214 [69].

SQSTM1 (also called p62) is a scaffold ubiquitin-binding protein that targets proteins for degradation by selective autophagy [138,139,140,141,142,143]. It is a 440 amino acid protein (Figure 2) with a N-terminal self-oligomerization domain (the PB1 domain) followed by a central zinc finger (ZZ) and a tumor necrosis factor receptor-associated factor 6 (TRAF6)-binding domain (TB) [144]. The TB domain functions as an activator of the NF-ĸB signaling pathway [145]. SQSTM1 exhibits a central binding domain for microtubule-associated proteins 1A/1B light chain 3B (LC3). The LC3-interacting domain (LIR) is essential for the induction of autophagy-related functions of SQSTM1, as well as for its own autophagic degradation [141]. The C-terminal of SQSTM1, the ubiquitin associated domain (UBA), binds to polyubiquitin chains and ubiquitinated proteins (www.uniprot.org, protein ID: Q13501) [144]. SQSTM1 is detected in cytoplasmic aggregates and so called aggresome-like induced structures (ALIS) [140]. ALIS are transient cytoplasmic aggregates that are generated in response to cellular stress. They serve as storage sites for polyubiquitinated proteins destined for proteasomal degradation [140,143]. The formation of ALIS is in fact dependent on SQSTM1. Mutation of either the PB1 or the UBA domain inhibited stress-induced ALIS formation and led to the spreading of SQSTM1 throughout the cytoplasm. Interestingly, overexpression of mutant SQSTM1 with disrupted UBA (SQSTM1_UBA_) domain was sufficient to abrogate ALIS formation, despite the presence of endogenous SQSTM1 [140]. In the SQSTM1-NUP214 fusion protein, the LIR and UBA domains of SQSTM1 are lacking, but in contrast to SQSTM1_UBA_, SQSTM1-NUP214 localized to both nuclear and cytoplasmic bodies [69,73].

Inhibition of CRM1 by leptomycin B (LMB) dissolved nuclear bodies containing SQSTM1-NUP214, but not the cytoplasmic ones, indicating that the molecular basis for the formation of nuclear and cytoplasmic bodies of SQSTM1-NUP214 is distinct. While nuclear SQSTM1-NUP214 bodies seem to depend on CRM1-NUP214_FG_ interaction, cytoplasmic bodies may result from the PB1-mediated oligomerization.

Recruitment of LC3 to autophagosomes is mediated by the LIR domain of SQSTM1 [141]. SQSTM1-NUP214 cytoplasmic bodies could recruit endogenous SQSTM1, but not LC3, which may result in hampered activation of autophagy. Furthermore, the absence of the LIR domain in SQSTM1-NUP214 may indicate that the fusion protein is more stable than wild type SQSTM1. In cells, the exclusion of LC3 from SQSTM1-NUP214 foci may enhance the stability of endogenous SQSTM1, due to impaired binding of LC3. SQSTM1 is a known activator of the NF-ĸB signaling pathway and this activity depends on the TB domain, which is conserved in SQSTM1-NUP214 [146]. Constitutive activation of NF-ĸB is observed in ALL patients [147]. Therefore, it is plausible that, by preserving the TB domain of SQSTM1 in SQSTM1-NUP214 and/or stabilizing normal SQSTM1, this fusion protein contributes to sustained NF-ĸB signaling in ALL.

## 8. Conclusions

Among the multiple causes of leukemia, chromosomal translocations of the *NUP214* gene may occur as a result of previous chemotherapy or as de novo genomic aberrations. NUP214 is part of the NPC and a critical player in the nuclear export of proteins and mRNA. Patients with *NUP214* rearrangements have an unfavorable prognosis due to rapid disease progression and ill treatment response. NUP214 fusion proteins typically preserve their FG domain, almost entirely or in part, indicating that the FG domain makes an important contribution to the transformation activity of NUP214 fusion proteins. At the molecular level, the preservation of NUP214_FG_ suggests that the fusion proteins influence nucleocytoplasmic transport. SET-NUP214, DEK-NUP214, and SQSTM1-NUP214 in fact all sequester CRM1 to foci in the nucleus. Another common feature of the three NUP214 chimera is the upregulation of clustered *HOX* genes, which likely contributes to the differentiation block observed in cells expressing these fusion proteins. In basal conditions, *HOX* genes are expressed at early stages of cellular maturation and their expression is progressively silenced during differentiation [148]. It is still unclear how NUP214 fusion proteins sustain *HOX* gene expression, but one can envision several scenarios: (1) defective nuclear protein export may result in the nuclear accumulation of transcription factors that promote *HOX* gene upregulation. (2) Spreading of histone hypoacetylation due to the preservation of the functional domains of SET and DEK in SET-NUP214 and DEK-NUP214, respectively. Deregulated HAT inhibition by SET or DEK may cause the silencing of genes necessary for hematopoietic differentiation (Figure 3). (3) Recruitment of additional factors to *HOX* promoters. Indeed, SET-NUP214 requires endogenous SET and DOT1L for the activation of *HOXA* genes. SET and DOT1L both act as activators of *HOX* gene expression, suggesting that SET-NUP214 stabilizes the association of SET and DOT1L with chromatin, which causes aberrant *HOX* gene activation. (4) Indirect activation of *HOX* genes by impaired cellular differentiation. DEK is known to cooperate with the myeloid transcription factor C/EBPα to promote myeloid differentiation of early hematopoietic progenitors. This interplay is disrupted by FLT3 and FLT3-ITD mediated phosphorylation of C/EBPα. One can envision that DEK-NUP214 can cooperate with either FLT3 or FLT3-ITD to promote the silencing of C/EBPα-target genes, which would result in impaired differentiation and could contribute to sustained upregulation of *HOX* genes (Figure 4).

The association of NUP214 fusion proteins with different leukemia subtypes suggests that distinct molecular mechanisms drive malignant transformation. SET-NUP214 and DEK-NUP214 patients exhibit different mutation profiles and gene expression patterns. The type of mutations associated with either SET-NUP214 or DEK-NUP214 might help to explain the observed differences in the course of the disease. Furthermore, the multiple functions of SET, DEK, and SQSTM1 add layers of complexity to the biological functions of NUP214 fusion proteins. SET and DEK are two chromatin remodeling proteins with multiple roles in transcriptional regulation. Both proteins are part of transcriptional repressive complexes that promote histone hypoacetylation. The two proteins regulate the status of chromatin relaxation by inhibiting the activity of the HATs p300/CBP and PCAF, but are paradoxically also essential for activation of gene transcription. How SET and DEK activity is exactly regulated is far from being understood.

The SQSTM1-NUP214 has been only detected in rare cases of ALL, and its role in cellular transformation is still unclear. SQSTM1 is a ubiquitin binding protein that targets proteins for degradation by autophagy. It is not known if SQSTM1-NUP214 affects autophagy in leukemia cells. Nevertheless, the observation that the fusion protein can partially co-localize with endogenous SQSTM1, but not with the autophagy marker LC3, suggests that SQSTM1-NUP214 might affect this process.The molecular mechanisms driving malignant transformation by NUP214 fusion proteins are far from being understood. Despite being a recurrent event, chromosomal translocations involving the *NUP214* locus account for rare leukemia cases, which hampers the access of researchers to patients’ samples. SET-NUP214 and DEK-NUP214 were first described more than 30 years ago, but only recently did research start to reveal the biological properties of these fusion proteins. The multiple functions of NUP214 fusion partners increase the complexity of potential roles of NUP214 fusion proteins in leukemia. It is therefore essential to understand how these proteins function in cells, and how deregulation of their functions may contribute to malignancy. This will be vital for the development of new treatment strategies for NUP214 patients that do not respond to current treatment options.

## Figures and Tables

**Figure 1 cells-08-00076-f001:**
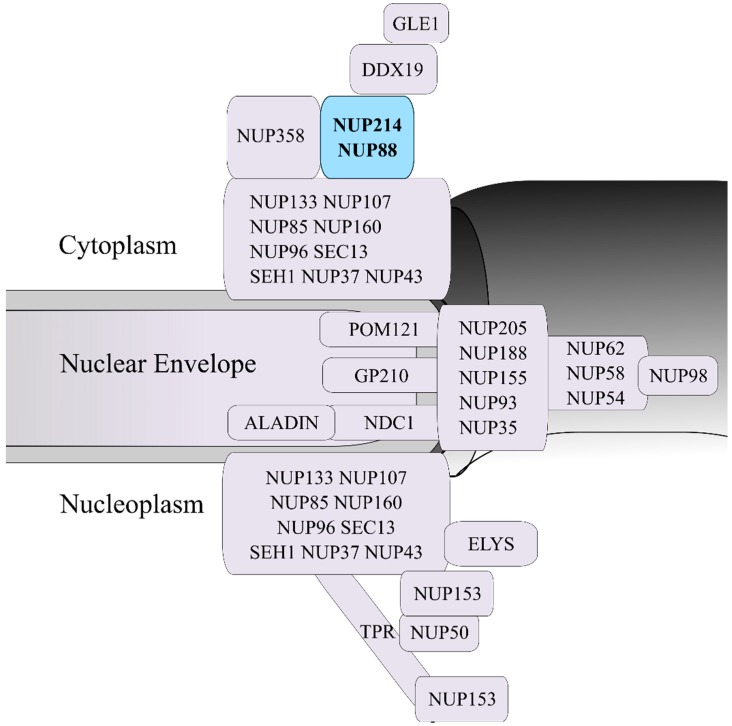
Schematic representation of the human nuclear pore complex (NPC) and the localization of nucleoporins within the NPC 3D structure. The NUP214-NUP88 complex, which is localized in the cytoplasmic side of the NPC, is highlighted in blue.

**Figure 2 cells-08-00076-f002:**
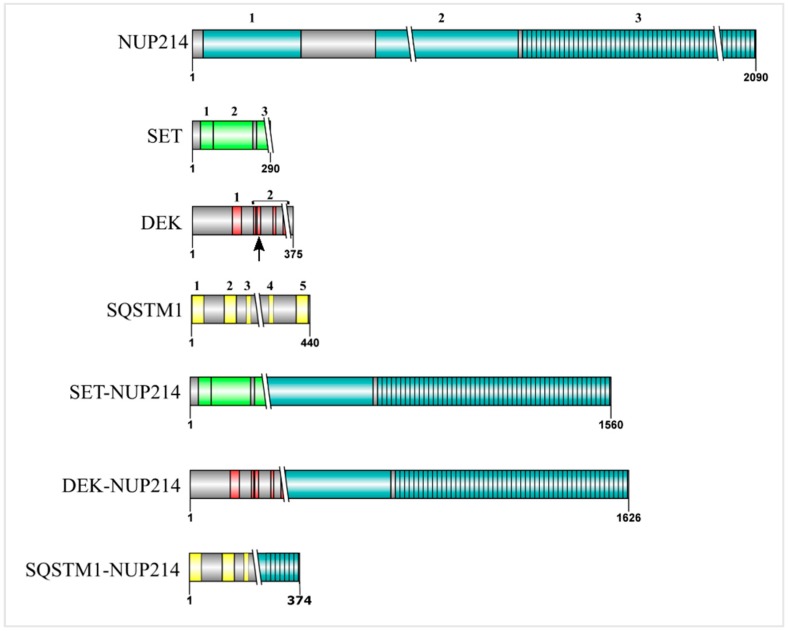
Schematic representation of NUP214 and its binding partners in leukemogenic NUP214 fusion proteins. The numbers indicate the specific domains of each protein. Crossing lines (\\) represent the breakpoints in the respective fusion protein. NUP214: 1—β propeller, 2—Coiled coil, 3—FxFG domain; SET: 1—dimerization domain, 2—earmuff domain, 3—acidic domain; DEK: 1—SAF-box domain, 2—acidic domains (overlaps with the second DNA binding domain, represented by the arrow); SQSTM1: 1—PB1 domain, 2—Zinc Finger, 3—TB domain, 4—LIR domain, 5—UBA domain.

**Figure 3 cells-08-00076-f003:**
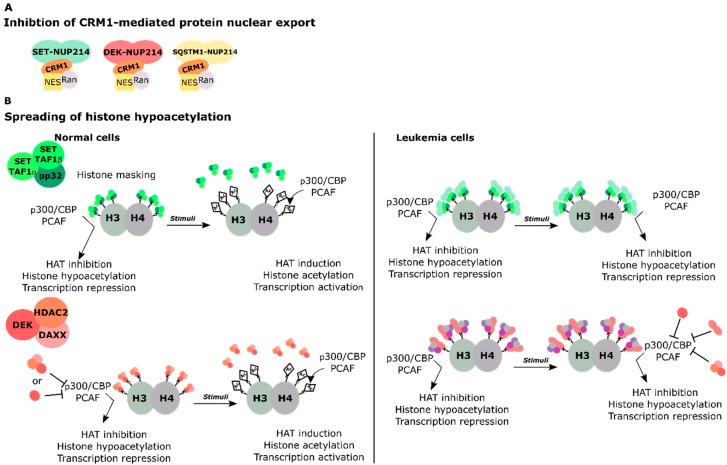
Putative models of gene regulation by SET-NUP214, DEK-NUP214, and SQSTM1-NUP214 fusion proteins. (**A**) Inhibition of CRM1-mediated nuclear export by sequestering nuclear export complexes into nuclear bodies formed by the fusion proteins. This may lead to nuclear accumulation of transcription factors that activate of *HOX* gene expression; (**B**) Spreading of histone hypoacetylation and transcriptional repression of genes involved in *HOX* activation. SET and DEK inhibit the transcriptional activity of the HATs p300/CBP and PCAF. Upon transcriptional stimuli, the interaction of SET and DEK with histone lysines is disrupted and HATs can co-activate transcription. In leukemia, SET-NUP214 preserves the dimerization domain of SET. SET-NUP214 may enhance the binding of endogenous SET to histones, in a mechanism similar to histone masking. DEK-NUP214 retains most of the histone binding domains of DEK and thus it may preserve histone binding capacity. In this model, DEK-NUP214 recruits additional, yet unknown, factors to chromatin. The NUP214_FG_ portion of the fusion proteins may contribute to aberrant histone hypoacetylation activity of SET and DEK due to their altered intracellular localization, which would result in the transcriptional repression of otherwise active genes.

**Figure 4 cells-08-00076-f004:**
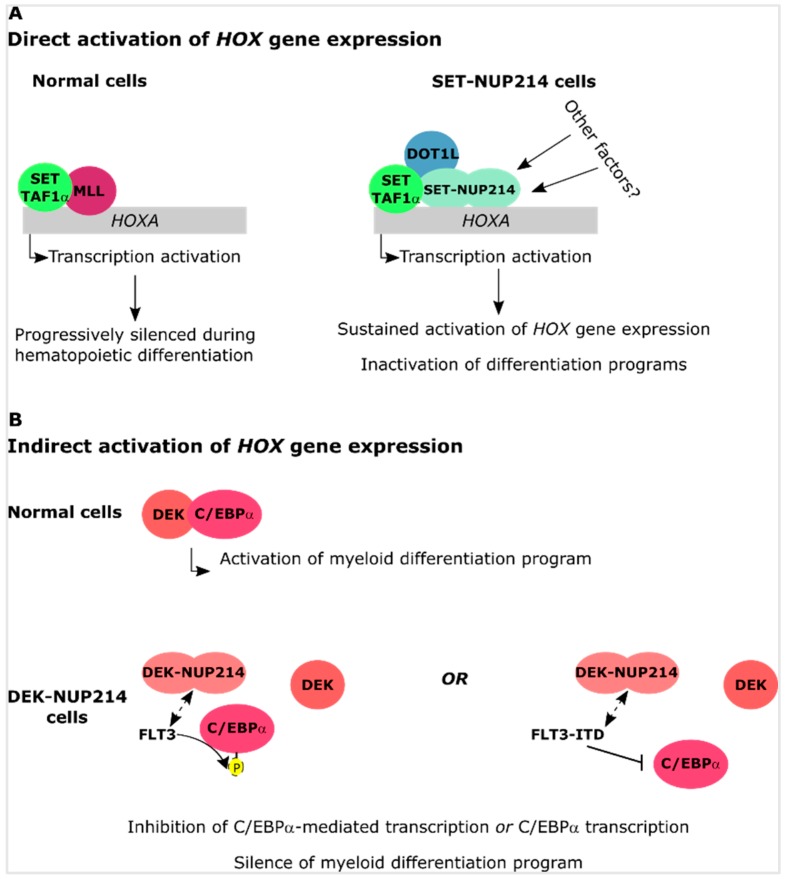
Potential mechanisms for upregulation of *HOX* genes by SET-NUP214 and DEK-NUP214. **(A**) In normal cells, SET acts jointly with MLL to promote *HOX* gene expression. *HOX* genes are progressively silenced during cell differentiation. In leukemia cells, SET-NUP214 recruits endogenous SET and DOT1L to maintain high HOX expression levels. Additional factors might be necessary to sustain *HOX* expression. (**B**) DEK and the myeloid transcription factor C/EBPα cooperate to promote C/EBPα-mediated myeloid differentiation. FLT3 and FLT3-ITD can disrupt the interaction between DEK and C/EBPα, either by phosphorylation or inhibition of C/EBPα transcription. DEK-NUP214 may cooperate with wild-type FLT3 to inhibit C/EBPα activity by increasing its phosphorylation. This might be further potentiated by DEK-NUP214.

**Table 1 cells-08-00076-t001:** Chromosomal translocations involving the *NUP214* gene and resulting fusion proteins reported in the literature.

Chromosomal Translocation	Fusion Protein	Leukemia Subtype	Refs.
**del9(9q34.11; 9q34.13)** **t(6;9) (q21;q34.1)**	SET-NUP214	T-ALL, ETP-ALL, AUL, AML	[65,66,67,72]
**t(6;9) (p23;q34)**	DEK-NUP214	AML	[65,66,67]
**der(5) t(5;9) (q35;q34)**	SQSTM1-NUP214	T-ALL	[69]
**9q34 episome amplification**	NUP214-ABL1	T-ALL	[68]

T-ALL: T-cell acute lymphoblastic leukemia; ETP-ALL: early thymic progenitor-ALL; AUL: acute undifferentiated leukemia; AML: acute myeloid leukemia.

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
