# Peer review of "NUP214 in Leukemia: It’s More than Transport"

_cells, 2019, doi:10.3390/cells8010076_

Reviewer 1 Report

This is a review that discuss about the involvement of nucleoporins, particularly NUP214, in leukemogenesis. The authors discuss about the normal biological functions of NUP214 and its fusion proteins (SET, DEK, and SQS-TM1) and the mechanisms of leukemogenesis mediated by NUP214 fusion proteins. Overall, the review has been written in a concise and comprehensive manner.

Minor:

p6 of 22 line 34-35  it needs to introduce those difference in phenotypes between transgenic mice and human.

p7 of 22 line 13-14 it needs a revision, it is not so easy to understand  two "this" exact meaning.

Author Response

“p6 of 22 line 34-35  it needs to introduce those difference in phenotypes between transgenic mice and human.”

We have rephrased the text and described the different phenotpyes more precisely.

“p7 of 22 line 13-14 it needs a revision, it is not so easy to understand  two "this" exact meaning.”

We agree with the Reviewer that the text was not exactly clear and have rephrased the sentence.

Reviewer 2 Report

There is a major mismatch with references and text. Authors may need to go through them and submit again for review. 

As after (Line 3) this (Ref 5) and many other references from here on are not matching with text. Authors may need to go through all the references and make sure they all match properly.

This is difficult to make the context of the text as the references are not matching so any further comments can be submitted later after corrected version will be submitted.

Author Response

We apologize for the mismatch of some references. We went through all references and corrected mismatching ones. The references that were corrected are highlighted in green.

Round  2

Reviewer 2 Report

In this review article, the authors elaborate on the role of the NUP214 in Leukemia. The primary mechanism which is Chromosomal translocations involving the NUP214 locus are recurrent in acute leukemia and frequently fuse the C-terminal region of NUP214 with SET and DEK, two chromatin remodeling proteins with roles in transcription regulation. SET-NUP214 and DEK-NUP214 fusion proteins disrupt protein nuclear export by inhibition of the nuclear export receptor CRM1, which results in the aberrant accumulation of CRM1 protein cargoes in the nucleus. SET-NUP214 is primarily associated with acute lymphoblastic leukemia (ALL), whereas DEK-NUP214 exclusively results in acute myeloid leukemia (AML), indicating different leukemogenic driver mechanisms.

Overall the review is well-written and elaborates the biological aspects on the role of Nup214 in Leukemia. The following concerns need to be addressed. Comments are as per revised manuscript I received (which does not include line numbers).

1)    In the introduction, it seems emphasis on selective transport is the only way a cargo can be transported through a nuclear pore. This is not the case. Non-selective transport also takes place for some cargoes which are smaller. The authors should discuss that.

2)    Sentence:” In this context, NUP214 is known to interact with both CRM1, the major exportin for proteins, and with nuclear RNA export factor 1 (NXF1), the principal mRNA export factor [27,28,58,59].” The references don’t make sense in the context of NXF1.

3)    "Nup214 fusion proteins preserve almost entire sequence SET & DEK." Please elaborate that how this can result in the deregulation of activity of SEC & DEK?

4)    The authors need to cite the paper which shows “DEK-NUP214 fusion protein preserves the acidic domains of DEK”.

5)    The authors stated that SQSTM1 is detected in cytoplasmic aggregates under cellular stress. Does the nuclear import of SQSTM1 get blocked under stress condition reported?

Author Response

1)    In the introduction, it seems emphasis on selective transport is the only way a cargo can be transported through a nuclear pore. This is not the case. Non-selective transport also takes place for some cargoes which are smaller. The authors should discuss that.

As suggested by the reviewer, we are now mentioning passive, non-selective transport through NPCs in the introduction (line 64-65).

2)    Sentence:” In this context, NUP214 is known to interact with both CRM1, the major exportin for proteins, and with nuclear RNA export factor 1 (NXF1), the principal mRNA export factor [27,28,58,59].” The references don’t make sense in the context of NXF1.

The wrong reference has been corrected (line 119, highlighted in yellow).

3)    "Nup214 fusion proteins preserve almost entire sequence SET & DEK." Please elaborate that how this can result in the deregulation of activity of SEC & DEK?

SET-NUP214 and DEK-NUP214 indeed maintain large part of the FG portion of NUP214 fused with almost the entire sequence of either SET or DEK. The NUP214 FG domain interacts with CRM1, and CRM1, beyond its role as nuclear export factor, was found at the promoters of clustered HOXgenes. This chromatin-bound CRM1 recruits the NUP98-HOXA9 fusion protein to HOXgene promoters, which results in the aberrant expression of these HOXgenes. As outlined in lines 177 onwards, we therefore hypothesize that a similar mechanism may occur in cells expressing either SET-NUP214 or DEK-NUP214. Due to the integrity of the functional domains of SET and DEK in the fusion proteins, it is therefore conceivable that SET and DEK exert their function at “abnormal” regions in the chromatin, which may lead to aberrant gene expression. We have elaborated this a bit in the revised manuscript (line 208). For SET-NUP214 moreover, as detailed in lines 273-281, the binding of SET-NUP214 to GR responsive elements (GREs) on the DNA is not disrupted by corticosteroids, in contrast to the binding of SET to GREs. Therefore, the DNA binding capacity of SET is maintained in SET-NUP214, the stability of the binding, however, is affected by the presence of the NUP214 portion and this may affect transcriptional regulation of GR target genes.

4)    The authors need to cite the paper which shows “DEK-NUP214 fusion protein preserves the acidic domains of DEK”.

The references has been added (line 167, 206, 320 highlighted in yellow). Ref 67 shows the DEK cDNA sequence and maps the breakpoint of DEK in DEK-NUP214, Ref 72 is the first description of SET-NUP214, Ref 117 maps the acidic domains of DEK.

5)    The authors stated that SQSTM1 is detected in cytoplasmic aggregates under cellular stress. Does the nuclear import of SQSTM1 get blocked under stress condition reported?

SQSTM1 is known to shuttle between the nucleus and the cytoplasm. Accordingly, classic NLS and NES signals have been mapped in SQTM1. Immunofluorescence studies have shown that SQSTM1 is predominantly localized to the cytoplasm, however there is little information about nuclear SQSTM1. A recent report has shown that SQSTM1 is important for the formation of aggresome-like structures called ALIS (Cabe et al., 2018). ALIS form under stress conditions and knock-down of SQSTM1 resulted in decreased ALIS formation Cellular stress was induced by exposing cells to lipopolysaccharide, however, at least to our knowledge, it is unclear whether this type of cellular stress affects the nuclear import of SQSTM1.

Round  3

Reviewer 2 Report

Authors have responded to all the issues satisfactorily and the article is suitable for the publication. This is recommended to keep references correctly organized during the process.